# The glycosyltransferase POGLUT1 regulates muscle stem cell development and maintenance in mice

Soomin Cho[1]◉, Emilia Servián-Morilla[2,3]◉, Victoria Navarro[2,3],
Beatriz Rodriguez-Gonzalez[2,3], Youxi Yuan[4], Raquel Cano[2,3], Arjun A. Rambhiya[4],
Radbod Darabi[5,6], Robert S. Haltiwanger[4], Carmen Paradas[2,3]*,
Hamed Jafar-Nejad[1,7,8]*

**1** Development, Disease Models & Therapeutics Graduate Program Baylor College of Medicine, Houston, Texas, United States of America, **2** Neuromuscular Disorders Unit, Department of Neurology, Instituto de Biomedicina de Sevilla, Hospital U. Virgen del Rocío/CSIC/Universidad de Sevilla, Sevilla, Spain, **3** Centro de Investigación Biomédica en Red sobre Enfermedades Neurodegenerativas (CIBERNED), Madrid, Spain, **4** Complex Carbohydrate Research Center, Department of Biochemistry and Molecular Biology, University of Georgia, Athens, Georgia, United States of America, **5** Institute of Muscle Biology and Cachexia, University of Houston, Houston, Texas, United States of America, **6** Department of Pharmacological and Pharmaceutical Sciences, University of Houston, Houston, Texas, United States of America, **7** Department of Molecular and Human Genetics, Baylor College of Medicine, Houston, Texas, United States of America, **8** Genetics and Genomics Graduate Program, Baylor College of Medicine, Houston, Texas, United States of America

◉ These authors contributed equally to this work.
* cparadas@us.es (CP), hamedj@bcm.edu (HJN)

## Abstract

Mutations in protein *O*-glucosyltransferase 1 (*POGLUT1*) cause a recessive limb-girdle muscular dystrophy (LGMDR21) with reduced satellite cell number and NOTCH1 signaling in adult patient muscles and impaired myogenic capacity of patient-derived muscle progenitors. However, the *in vivo* roles of POGLUT1 in the development, function, and maintenance of satellite cells are not well understood. Here, we show that conditional deletion of mouse *Poglut1* in myogenic progenitors leads to early lethality, postnatal muscle growth defects, reduced *Pax7* expression, abnormality in muscle extracellular matrix, and impaired muscle repair. *Poglut1*-deficient muscle progenitors exhibit reduced proliferation, enhanced differentiation, and accelerated fusion into myofibers. Inducible loss of *Poglut1* in adult satellite cells leads to their loss of quiescence and precocious differentiation, and impairs muscle repair upon serial injury. Cell-based signaling assays and mass spectrometric analysis indicate that POGLUT1 is required for the activation of NOTCH1, NOTCH2, and NOTCH3 in myoblasts and that NOTCH3 is a target of POGLUT1 like NOTCH1 and NOTCH2. These observations provide insight into the roles of POGLUT1 in muscle development and repair and the pathophysiology of LGMDR21.

**Data availability statement:** The authors confirm that all data underlying the findings are fully available without restriction. All relevant data are within the paper and its Supporting Information files.

**Funding:** This work was supported by the National Institute of Arthritis and Musculoskeletal and Skin Diseases (NIH R01AR076770 to HJN and RD), by the Instituto de Salud Carlos III (grant numbers PI21/00759 to ESM and PI19/01497 to CP), by the Junta de Andalucia, Consejería de Salud y Consumo (grant number PIER-0100-2019 to ESM), by Fundación Feder (grant number AI-2022-020 to CP), and by the Eunice Kennedy Shriver National Institute of Child Health and Human Development to Baylor College of Medicine IDDRC (P50 HD103555). The funders had no role in study design, data collection and analysis, decision to publish, or preparation of the manuscript.

**Competing interests:** The authors have declared that no competing interests exist.

## Author summary

Skeletal muscle growth and repair in vertebrates depend on specialized cells expressing the transcription factor PAX7. During development, these progenitor cells drive muscle growth, with a subset later becoming adult muscle stem cells—known as satellite cells—which usually remain dormant until muscle injury occurs. Upon activation, some satellite cells differentiate to repair damage, while others return to dormancy to maintain long-term regenerative capacity. Using mouse models, here we report that the enzyme POGLUT1 plays an essential role in PAX7-expressing cells at late embryonic and early postnatal stages to ensure proper muscle growth and satellite cell establishment. POGLUT1 is also required for the maintenance of adult satellite cells and their return to dormancy after injury repair. Our data indicate that POGLUT1 regulates these processes by promoting Notch signaling, which itself induces PAX7 expression, and we demonstrate that this requirement extends across all three Notch receptors expressed in muscle progenitors. These findings help explain how pathogenic *POGLUT1* variants cause muscular dystrophy in humans and identify POGLUT1 as a critical regulator of muscle stem cell biology throughout life.

## Introduction

Muscle satellite cells (SCs) are essential for muscle development, maintenance, and repair [1]. These cells reside between the basal lamina and sarcolemma of muscle fibers in a quiescent state until they are activated. Once activated, they proliferate and form myoblasts, which differentiate and either fuse to one another to form new myofibers or fuse to existing myofibers to promote muscle growth or to repair muscle injury [2]. The Notch signaling pathway plays a pivotal role in maintaining the quiescence and promoting the activation of muscle satellite cells. Notch signaling also regulates the balance between self-renewal and differentiation of muscle stem cells [3]. Loss of function studies for key components of the Notch signaling pathway including *Rbpj* and *Dll1* indicated that disruption of this pathway during muscle development leads to premature differentiation and depletion of muscle progenitor cells [4,5]. In addition, multiple studies have shown that Notch signaling is required in adult mice to prevent the spontaneous or premature differentiation of satellite cells and to maintain a muscle stem cell pool capable of repairing muscle damage [6–9].

We have previously identified a new form of muscular dystrophy called limb-girdle muscular dystrophy autosomal recessive 21 (LGMDR21), which is caused by pathogenic variants in the *POGLUT1* (Protein *O*-Glucosyltransferase 1) gene [10,11]. In most patients, the disease manifests as adult-onset, although one case of congenital and two cases of infantile onset have been reported as well [10,11]. POGLUT1 is an enzyme involved in the post-translational modification of proteins, including Notch receptors and ligands [12]. Specifically, it adds *O*-linked glucose to a conserved consensus sequence within the epidermal growth factor-like (EGF) repeats of its

target proteins [13]. Biochemical experiments and cross-species rescue experiments in transgenic *Drosophila* indicate the *POGLUT1* variants identified in LGMDR21 patients significantly reduce its enzymatic activity [10,11]. Patients with LGMDR21 exhibit progressive muscle weakness, accompanied by reduced NOTCH1 signaling and a decrease in the number of satellite cells marked by PAX7 expression [10,11]. Moreover, we recently reported that myoblasts generated from induced pluripotent stems cells (iPSCs) derived from an LGMDR21 patient showed impaired myogenesis in cell culture experiments and defective *in vivo* engraftment and mislocalization of engrafted satellite cells in mice, all of which were rescued upon CRISPR-mediated *POGLUT1* gene correction in patient iPSCs [14]. Together, these studies highlight the important role of POGLUT1 in muscle health and suggest a key role for this enzyme in PAX7+ satellite cells. However, the roles of POGLUT1 in the development, function, and maintenance of muscle progenitors in an *in vivo* mammalian model system is not well understood.

To fill this gap in knowledge, we have generated a mouse model with conditional knockout (cKO) of *Poglut1* in muscle progenitors using *Pax7^Cre* [15], as germline deletion of *Poglut1* results in embryonic lethality [16,17]. *Poglut1-cKO* mice exhibit severe postnatal growth defects specifically affecting the muscle, with most animals succumbing to lethality within the first month of life. The mutant mice display a reduction in the number of PAX7+ cells, show *in vivo* and *ex vivo* evidence for reduced proliferation and precocious fusion of myoblasts, and exhibit an impairment in muscle repair upon injury. Experiments using inducible *Pax7-Cre-ERT2* mice reveal that POGLUT1 is also essential for maintaining satellite cell quiescence and repairing muscle damage in adult mice. Cell-based signaling assays indicate that POGLUT1 promotes signaling mediated by all three Notch receptors involved in satellite cell development and function (i.e., NOTCH1, 2 and 3). Altogether, our data establish a critical role for *Poglut1* during myogenesis and for the maintenance of adult satellite cells.

## Results

### Loss of *Poglut1* with *Pax7-Cre* leads to postnatal growth retardation, muscle weakness, and early lethality

To study the role of POGLUT1 in developing muscle progenitor cells, we generated *Poglut1* conditional knockout mice by crossing the *Poglut1^flox* allele [18] with Pax7^Cre [15]. The mutant mice, hereafter called *Poglut1-cKO*, were not distinguishable from the control littermates at the time of birth (Fig 1A). However, we observed postnatal growth retardation in the mutant mice by postnatal day 14 (P14-P21) (Fig 1A). To confirm the knockout of *Poglut1*, we performed western blot assays and observed a decrease in POGLUT1 level in both isolated satellite cells and whole muscle lysate from *Poglut1-cKO* forelimb and hindlimb muscles compared to sibling controls (Fig 1B and 1C). TA muscle weight is strongly reduced in P21 cKO animals compared to controls (Fig 1D). Importantly, the cKO animals did not show a statistically significant reduction in tibia length, and even when normalized by tibia length, the average TA weight in P21 cKO mice still showed ~55% reduction compared to controls on average, indicating muscle-specific growth defects (Fig 1D). The mutant mice also had a much shorter lifespan compared to the controls. When fed on a regular diet, they could not survive past the first month of life (Fig 1E). Feeding soft food allowed some *Poglut1-cKO* animals to survive longer (Fig 1E). However, even upon feeding on soft food the mutant mice had significantly lower weights compared to sibling controls (Fig 1F). Moreover, the *Poglut1-cKO* animals' maximum speed in open field experiments and their grip strength were significantly reduced compared to sibling controls, suggesting reduced muscle strength in the mutant mice (Fig 1G and 1H). Of note, when normalized by body weight, grip strength did not differ significantly between *Poglut1-cKO* and control animals, suggesting that the observed reduction in absolute grip strength in mutants can be attributed to their reduced body weight (S1A Fig). These behavioral deficits were accompanied by a significant reduction in the postsynaptic area in the neuromuscular junctions (NMJs) of *Poglut1-cKO* animals (Fig 1I and 1J). Electrophysiological recording of the mutant and control levator auris longus (LAL) muscle showed that while the miniature (mEPPs) and evoked end-plate potentials (EPPs) in *Poglut1-cKO* NMJs were comparable to those in control NMJs, synaptic vesicles in the NMJs of *Poglut1-cKO*

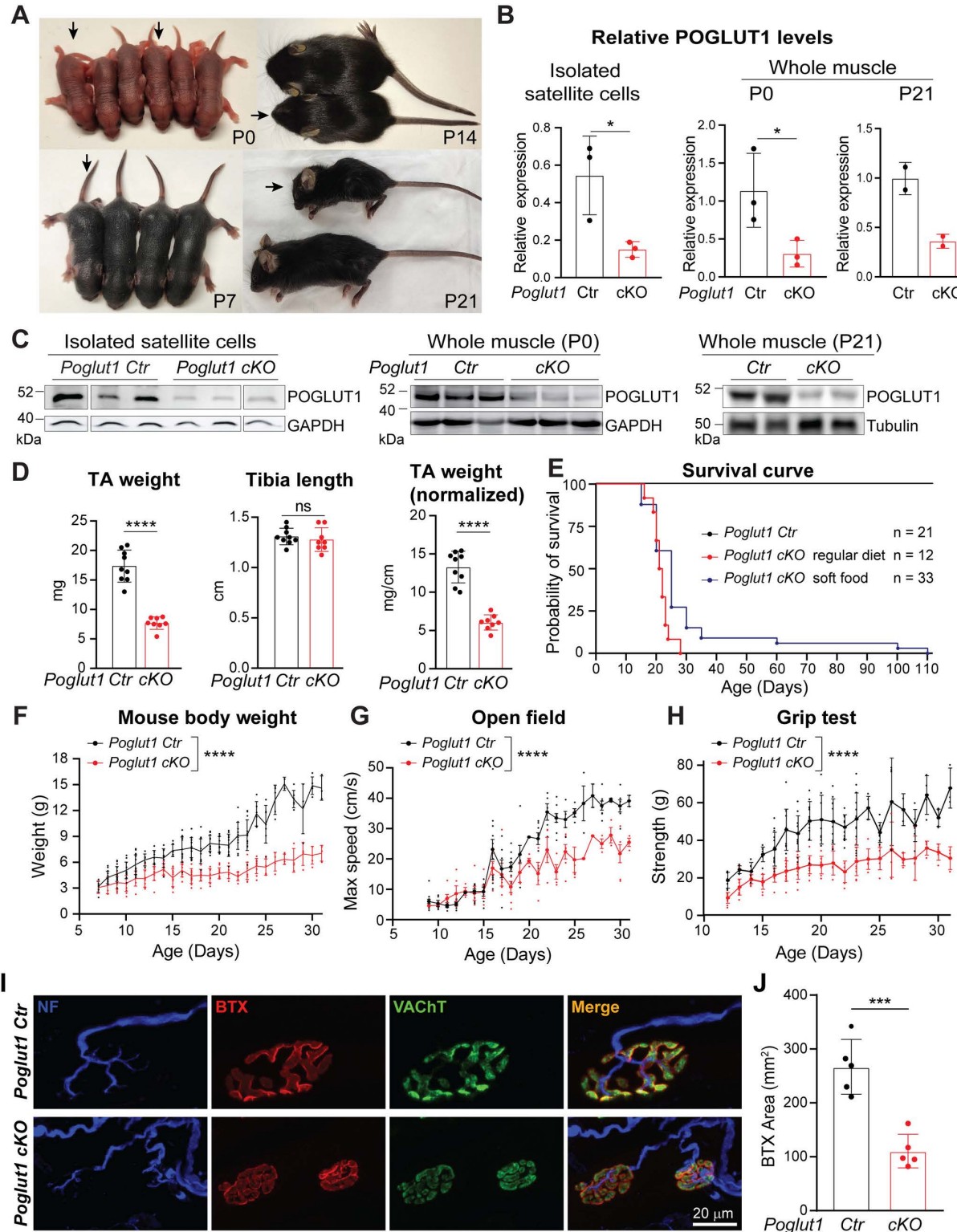

**Fig 1. Loss of *Poglut1* with *Pax7-Cre* leads to postnatal growth retardation, early lethality, muscle weakness, and impaired NMJ formation. A)** Images of *Poglut1-cKO* mouse (arrows) and their littermate controls at P0, P7, P14 and P21. **B, C)** Western blot images and quantification of relative levels of POGLUT1 from isolated satellite cell lysates (B, left panel) and whole muscle lysates (B, right panel) from forelimb and hindlimb muscles of

*Poglut1-cKO* and sibling controls. Anti-GAPDH and anti-tubulin blots are shown as loading controls. Each western blot lane and each circle in the graphs represents an individual animal. **D)** TA muscle weights, tibia lengths, and TA muscle weights normalized by tibia length of each mouse. **E)** Survival curves of *Poglut1-cKO* and sibling control mice with regular diet and with soft food. n = number of animals. **F–H)** Body weights **(F)**, open field test **(G)** and grip test **(H)** of control and *cKO* mice, with each dot representing an individual animal. **I)** Representative en face views of NMJs from the LAL muscle stained with BTX-Rho (red), which binds specifically to postsynaptic AChRs, anti-VAChT (green), which labels synaptic vesicles; and anti-neurofilament, which labels motor neuron axons (blue). Terminals of control and cKO mice at P20 are shown. **J)** Quantification of the postsynaptic areas in cKO and control terminals, in ages between P20 and P50 (*Poglut1-Ctr*: 5 mice (50 terminals); *Poglut1-cKO*: 5 mice (47 terminals)). Siblings without *Poglut1* deletion (*Poglut1^{+/+}*, *Poglut1^{flox/flox}*, or *Pax7^{Cre}*; *Poglut1^{+/+}*) were used as controls in all experiments. Mean±SD. Two-way ANOVA with Šidák's multiple comparisons test **(F–H)**; unpaired *t* test (B, D, **J**). Row factor (the overall effect of genotype) is shown for F–H. ns: not significant, *$P < 0.05$, ***$P < 0.001$, ****$P < 0.0001$.

LAL muscles showed a significantly reduced evoked neurotransmitter release (quantal content) compared to controls (S1B and S1C Fig). Since the *Pax7-Cre* strain used in our study is not reported to induce recombination in motor neurons [19], this presynaptic NMJ defect might be secondary to defects in postsynaptic NMJ abnormalities. Taken together, these results indicate that *Poglut1-cKO* mice exhibit postnatal muscle growth retardation accompanied by muscle weakness and NMJ abnormalities.

### Loss of *Poglut1* with *Pax7-Cre* leads to a severe reduction in the number of PAX7^+ cells, accompanied by abnormalities in satellite cell niche formation

Given the severe reduction in the number of PAX7^+ satellite cells in LGMDR21 [10,11], we sought to determine the impact of loss of *Poglut1* on muscle progenitor cells. To this end, we performed immunofluorescence studies on *Poglut1-cKO* and control TA muscles at different developmental stages. As expected, control muscles harbored many PAX7^+ cells at embryonic day 16 (E16), which were gradually reduced in numbers from E16 to P21 (Fig 2A). The total number of PAX7^+ cells was significantly reduced in mutant mice at all three time points (Fig 2A and 2B). qRT-PCR experiments showed that *Pax7* mRNA levels were strongly decreased in P0 and P21 *Poglut1-cKO* muscles (Fig 2C) and in satellite cells isolated from *Poglut1-cKO* muscles (Fig 2D), suggesting that the reduction in PAX7 in mutant muscle occurs at a transcriptional level.

We also studied the organization of the extracellular matrix (ECM) in mutant and control muscles and noticed an abnormal expression pattern for laminin in the mutant muscle (Fig 2A, asterisks). At P0, unlike control muscle fibers which were surrounded by strong laminin staining in most cases, some mutant muscle fibers showed very weak laminin staining at their junction with neighboring muscle fibers at this age (Fig 2A, asterisks). A closer inspection of E16 TA sections co-stained for laminin and collagen VI (ColVI; [20]), another component of the extracellular matrix, indicated that at this age, many of the junctions between neighboring myofibers exhibit weak laminin and ColVI staining in both *Poglut1-cKO* and control muscles (Fig 2E, asterisks). By P0, rather strong laminin and ColVI staining is observed around most myofibers in control muscles but weak staining for these markers persists in *Poglut1-cKO* muscles (Fig 2E). Of note, by P21, this phenotype is resolved in the mutant muscle (Fig 2A and 2E). Given these data and our previous observations on hypoglycosylation and reduced laminin binding capacity of α-dystroglycan in patient muscles, we examined the glycosylation status of α-dystroglycan and its laminin-binding ability in *Poglut1-cKO* and control muscles. In both P0 and P21 muscles, we found α-dystroglycan hypoglycosylation and reduced laminin binding (S2 Fig), further suggesting ECM defects.

To better characterize the effects of loss of *Poglut1* on satellite cell development in mice, we double stained TA sections with antibodies against PAX7 and another reliable marker for satellite cells, namely M-Cadherin [21,22]. M-Cadherin has shown to be implicated in the correct adhesion of SC to myofibers [23]. Interestingly, some PAX7^+ cells did not express M-Cadherin and vice versa (Fig 2F). Notably, quantification of single- and double-positive cells at P0 and P21 showed that only the PAX7, M-Cadherin double-positive cells were significantly reduced in mutant mice (Fig 2G). Overall, these data indicate that loss of *Poglut1* causes a severe decrease in the number of PAX7^+ cells starting at late embryonic stages, accompanied by abnormalities in the formation or maturation of the skeletal muscle extracellular matrix.

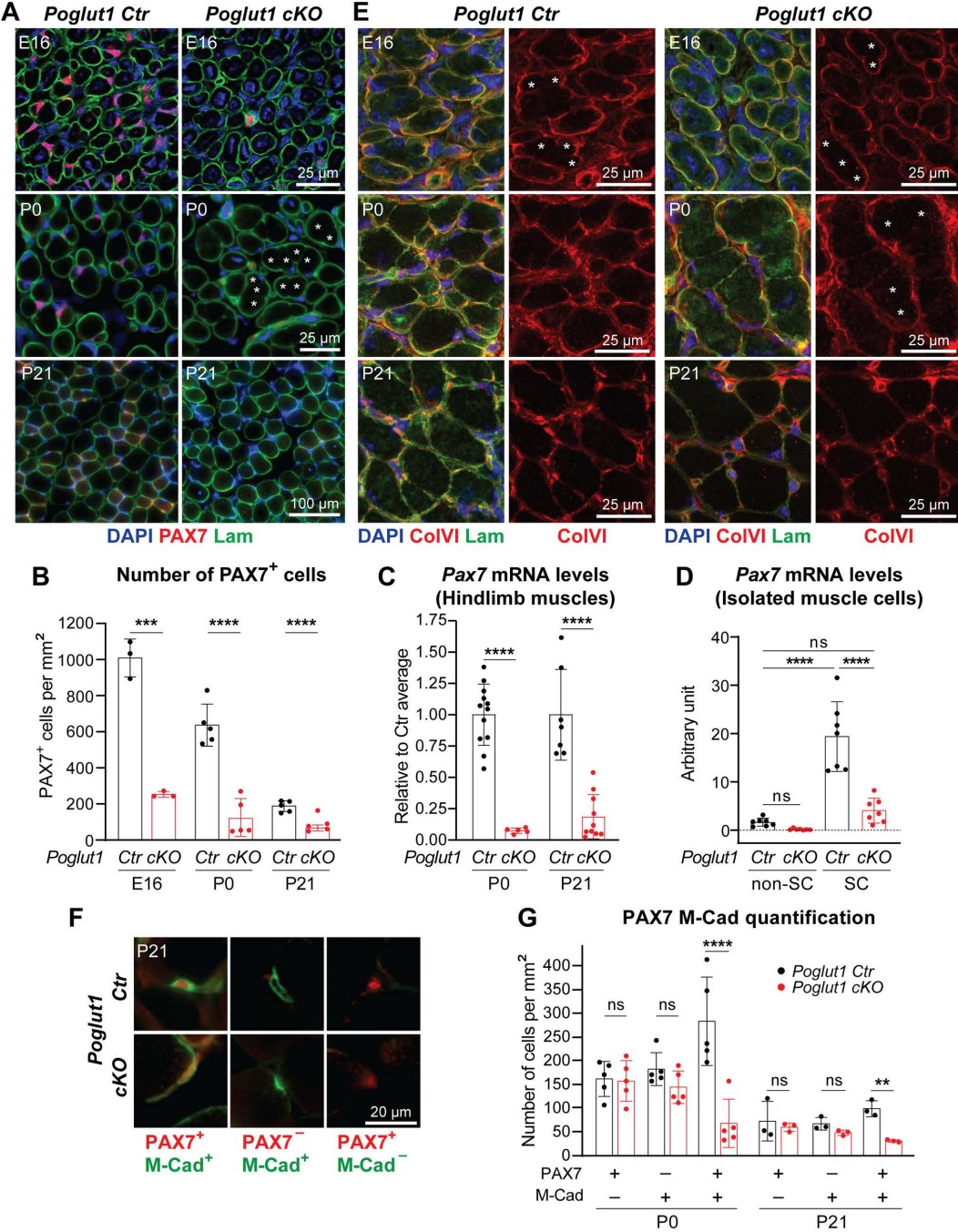

**Fig 2. Loss of *Poglut1* with *Pax7-Cre* leads to a severe reduction in the number of PAX7+ cells, accompanied by abnormalities in satellite cell niche formation. A)** Images of TA muscle sections from mice harvested at E16, P0, and P21and stained for DAPI (blue), Laminin (green), and PAX7 (red). Asterisks mark myofibers separated by very weak Laminin staining. **B)** Quantification of PAX7+ cells per mm² in sections from TA muscles harvested at E16, P0, and P21 from the indicated genotypes. **C)** *Pax7* mRNA levels measured by qRT-PCR from limb muscles of control and cKO mice. **D)** *Pax7* mRNA levels measured by qRT-PCR from satellite cells (SC) and other cells (non-SC) isolated from muscles of P4 control and cKO mice. **E)** Images of muscle sections from mice harvested at E16 (hindlimb), P0 (hindlimb), and P21 (TA) and stained for DAPI (blue), Laminin (green), and Collagen VI (red). Asterisks mark myofibers separated by very weak Laminin/ColVI staining. **F–G)** Representative images of M-Cadherin and PAX7 staining of TA muscle sections (F) along with the quantification of single/double positive cells per mm² (G). In B–D and G, each dot represents an animal. Siblings without *Poglut1* deletion (*Poglut1*+/+, *Poglut1*flox/flox, or *Pax7*Cre; *Poglut1*+/+) were used as controls in all experiments. Mean±SD is shown. Unpaired *t* test (B, C, **G**); one-way ANOVA with Tukey's multiple comparisons tests **(D)**. ns: not significant, **P < 0.01, ***P < 0.001, ****P < 0.0001.

**Poglut1-cKO muscle progenitors exhibit impaired proliferation, enhanced differentiation and accelerated fusion**

To determine the mechanism for the observed reduction in PAX7+ myogenic progenitors in *Poglut1-cKO* muscle, we examined whether myoblast differentiation and/or fusion was affected upon loss of *Poglut1.* We first quantified the ratio of myofibers with internal nuclei, recently reported to specifically result from the fusion of embryonic myogenic cells during limb myogenesis and also driven by myocyte-myocyte fusion in the first phase of postnatal muscle regeneration [24]. At E16, ~20% of myofibers had internal nuclei in both mutant and control TA muscles (Figs 3A and S3). At P0 and P21, only 2% and 1% of TA myofibers had internal nuclei in control muscles, respectively (Figs 3A and S3). In contrast, although the percentage of myofibers with internal nuclei was also reduced in mutant mice at P0 and P21 compared to E16, the mutant muscles had significantly higher ratios of myofibers with internal nuclei compared to control muscles at the same age (Figs 3A and S3). These data suggest enhanced differentiation of *Poglut1*-deficient progenitors into myogenic cells followed by continued fusion of these myogenic cells and/or a delay in the peripheral migration of internal nuclei, which normally occurs in the perinatal period [24]. We also observed an increase in the myofiber cross-sectional area (CSA) in mutant mice at P0 (Fig 3B), suggesting precocious fusion of *Poglut1*-deficient myoblasts into existing fibers in late embryonic/early postnatal stages. Of note, by P21, the average CSA in mutant TA muscles was significantly less than that in control TA muscles (Fig 3B). Moreover, the total number of myonuclei in extensor digitorum longus (EDL) myofibers of P21 mutant animals was less than half of that in control EDL myofibers (Fig 3C). These data suggest a decrease in the myogenic progenitor pool in *Poglut1-cKO* muscles, potentially due to the observed precocious differentiation and fusion. We also stained the control and mutant muscles at P21 for embryonic myosin heavy chain (eMHC), a form of myosin that is expressed in embryonic myofibers but disappears in the early postnatal period in most muscles and is not expressed in adult muscles unless there is muscle regeneration [25]. As expected, at P21 the control muscles did not show any embryonic myosin heavy chain (eMHC) expression (Fig 3D). However, many *Poglut1-cKO* myofibers were still positive for eMHC at this age (Fig 3D). This observation suggests that either some myofibers in the mutant mice are still immature and continue to express eMHC, or that there is an ongoing differentiation and fusion of the remaining myogenic progenitors due to the loss of *Poglut1* in PAX7+ cells (or both).

To directly assess the effects of loss of *Poglut1* on the behavior of PAX7+ cells, we isolated myogenic progenitors from P4 *Poglut1-cKO* and control muscles and cultured them under proliferation and differentiation conditions. This time-point falls in the expansion phase of PAX7+ cells (P0–P5), during which these cells are shown to have a high propensity for expansion and a relatively low tendency for differentiation and fusion [26]. As shown in Fig 3E and 3F, after 3–5 days in proliferation medium, mutant myoblast cultures had a significantly smaller number of cells compared to control cultures, even though all cultures were started with the same number of cells. These data suggest reduced proliferation of myogenic cells upon loss of *Poglut1,* although we cannot exclude a potential contribution from cell death, which was not directly assessed. In addition, the ratio of PAX7+ cells in the mutant myoblast culture was significantly decreased at all time points (Fig 3G). To assess the role of *Poglut1* in maintaining the quiescent state in PAX7+ muscle stem cells, we performed PAX7 MYOD Ki67 triple staining on *cKO* and control myoblast cultures at days 1, 3 and 5 and quantified the ratio of various cell states at each time point based on the expression of these markers. We found a significant reduction in the percentage of quiescent muscle stem cells (PAX7+ MYOD− Ki67−) in *Poglut1-cKO* cultures at days 1 and 3 compared to control cultures (Figs 3H and S4). Moreover, on day 5 the mutant cultures showed a strong and statistically significant increase in the percentage of non-cycling precursor cells (Fig 3H; PAX7− MYOD+ Ki67−). These observations indicate that loss of *Poglut1* impairs the ability of muscle stem cells to remain in a quiescent state and suggest that the mutant myogenic progenitors might undergo premature differentiation. To directly assess the effects of loss of *Poglut1* on the differentiation of myogenic progenitors, we switched confluent cultures of each genotype to differentiation medium and analyzed them after three days of culture in this condition. As shown in Fig 3I, 3J and 3K, *Poglut1-cKO* myoblast cultures harbored a larger number of differentiated myotubes, as evidenced by fusion index and the ratio of Myogenin+ cells. Together, these *in vivo* and *ex vivo* experiments indicate reduced proliferation and enhanced differentiation and fusion of mouse myoblasts upon loss of *Poglut1*.

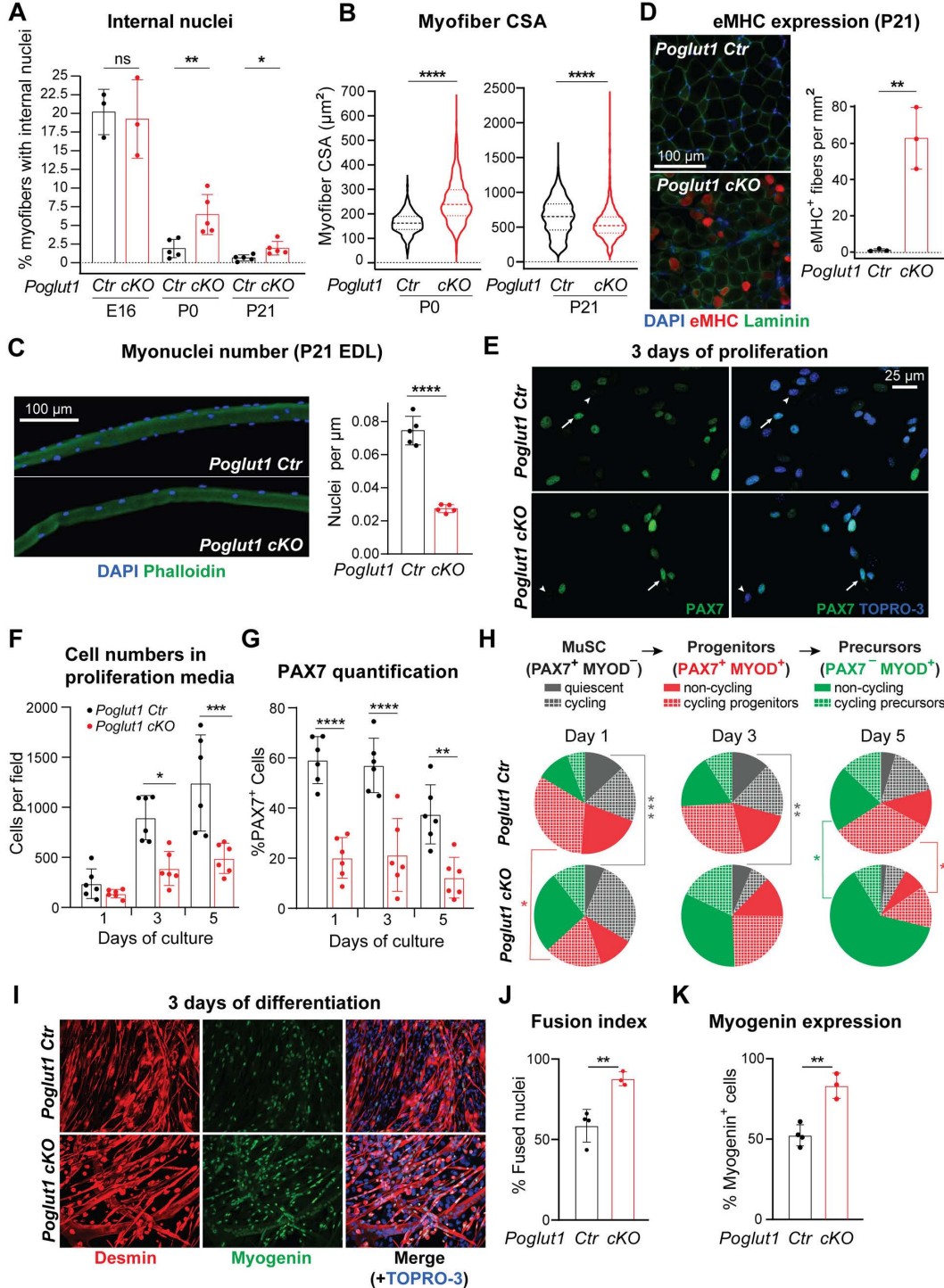

**Fig 3. Loss of *Poglut1* with *Pax7-Cre* leads to enhanced differentiation and accelerated fusion of muscle progenitors. A)** Quantification of percentage of myofibers with internal nuclei in TA muscles from the indicated genotypes at E16, P0, and P21. **B)** Violin plots of myofiber cross-sectional area (CSA) from P0 and P21 mice of the indicated genotypes. Horizontal lines indicate median ± one quartile. **C)** Immunofluorescence image of isolated myofibers of P21 mice stained with DAPI (blue) and Phalloidin (green) with quantification of the number of nuclei per μm of fiber length. **D)** Immunofluorescence images of TA cross-sections stained for DAPI (blue), eMHC (red), and Laminin (green), with quantification of eMHC+ fibers per mm² of section area. **E)** Immunofluorescence images of myoblasts stained for PAX7 (green) and the nuclei (TOPRO-3, blue). **F)** Quantification of primary myoblasts

from control and cKO mice (proliferation rate) at 1, 3 and 5 days of culture in proliferation medium. Note that the same number of cells was plated at the start of each culture to ensure comparability. **G)** Percentage of PAX7+ cells at 1, 3 and 5 days of proliferation. **H)** Pie charts showing the distribution of the myogenic populations based on their PAX7 and MYOD expression and cycling state (tracked by Ki67) on 1, 3, and 5 days of culture in proliferation medium. **I)** Immunofluorescence images of myoblasts stained for desmin (red) and myogenin (green) after three days of culture in differentiation medium. Nuclei were counterstained with TOPRO-3 (blue). **J)** Fusion index for cultures from panel I, defined as the percentage of nuclei in myotubes relative to the total number of nuclei in myogenic cells. **K)** Quantification of myogenin expression from indicated genotypes. In all graphs, each dot represents data from one animal. Siblings without *Poglut1* deletion (*Poglut1*+/+, *Poglut1*flox/flox, or *Pax7*Cre; *Poglut1*+/+) were used as controls in all experiments. Mean±SD. One-way ANOVA with Tukey's multiple comparisons test (F, G, H) or unpaired *t* test (A–D, J, K). ns: not significant, *$P < 0.05$, **$P < 0.01$, ***$P < 0.001$, and ****$P < 0.0001$.

### *Poglut1*-deficient muscles do not exhibit proper satellite cell activation and muscle repair upon muscle injury

PAX7+ quiescent satellite cells become activated upon nearby muscle injury and are essential for muscle repair [27]. While the mutant mice have much fewer PAX7+ cells compared to the controls, it was not known whether the remaining PAX7+ cells can function as satellite cells upon injury. To address this question, we induced muscle injury in mutant and control mice by injecting cardiotoxin (CTX) into TA muscles at P21 (Fig 4A), an age at which we consistently obtain *Poglut1-cKO* animals without any dietary changes (Fig 1E). Importantly, in WT C57BL/6 mice, 51% of PAX7+ cells are reported to be in the quiescent state at P21 [26] and the TA muscles of P21 mice exhibit a robust regenerative response to cardiotoxin-induced injury [28]. The injured tissue was harvested 5 and 14 days post injury (dpi). In control mice, at the site of CTX injury the myofibers showed strong eMHC staining and internal nuclei at five dpi (Fig 4A), both of which are observed during muscle regeneration upon injury [24,29,30]. By 14 dpi, the injured muscle showed robust repair and did no longer exhibit eMHC staining (Fig 4A). In contrast, the mutant mice did not show any signs of repair at these two timepoints (Fig 4A). These observations indicate that expression of *Poglut1* in muscle progenitors is essential for muscle repair in young adult mice and that the remaining PAX7+ cells in *Poglut1-cKO* TA muscle are not capable of muscle regeneration.

To further evaluate the functionality of PAX7+ cells in *Poglut1-cKO* muscles, we performed immunofluorescent staining on myofibers isolated from mutant and control mice. In agreement with our observation from muscle staining shown in Fig 2, the mutant animals showed a significant decrease in the number of PAX7+ cells per isolated myofibers (Fig 4B). The isolated fibers were then cultured for 48 hours post isolation as another form of injury to activate the satellite cells. Staining with antibodies against PAX7, MYOD and Ki67 indicated that each control myofiber has an average of two clusters (aggregate of 3 or more cells) composed of proliferating myogenic cells (Fig 4C–4E). In contrast, there were no clusters in the mutant fibers, most fibers lacked PAX7+ cells at this time, and only some fibers harbored MYOD+ single cells (Fig 4C–4E). These observations suggest that the remaining PAX7+ cells in the *Poglut1-cKO* TA muscle fail to proliferate upon muscle injury.

### *Poglut1* is required for the maintenance of the quiescent adult satellite cells and their self-renewal after cardiotoxin injury

To examine whether POGLUT1 is also required in adult satellite cells independent of its role during development, we crossed the *Poglut1*flox mice to *Pax7*CreERT2 [31] in order to generate inducible conditional knockout mice for this gene (*Poglut1 i-cKO*). *The i-*cKO animals also had a *ROSA-CAG-lox-stop-lox-tdTomato* allele [32] to mark the recombination in Cre+ cells. We induced recombination by five days of tamoxifen (TAM) injection and then injected CTX into the TA muscle on one side to induce injury, as shown in Fig 5Ai. In two additional cohorts, TAM injections and CTX injury were repeated once or twice on the same TA muscle with 25-day intervals before harvesting the tissue (Fig 5Aii and 5Aiii). As control, we used animals harboring both *Pax7*CreERT2 and *ROSA-CAG-lox-stop-lox-tdTomato* but lacking *Poglut1*flox and injected with the same TAM and CTX at the same timeline as the *i-cKO* mice.

Analysis of tdTomato and WGA (cell membrane marker) expression in injured TA muscles at 14 dpi showed that *i-cKO* muscles were efficiently repaired after one round of TAM injection and CTX injury, similar to control mice (Fig 5Bi).

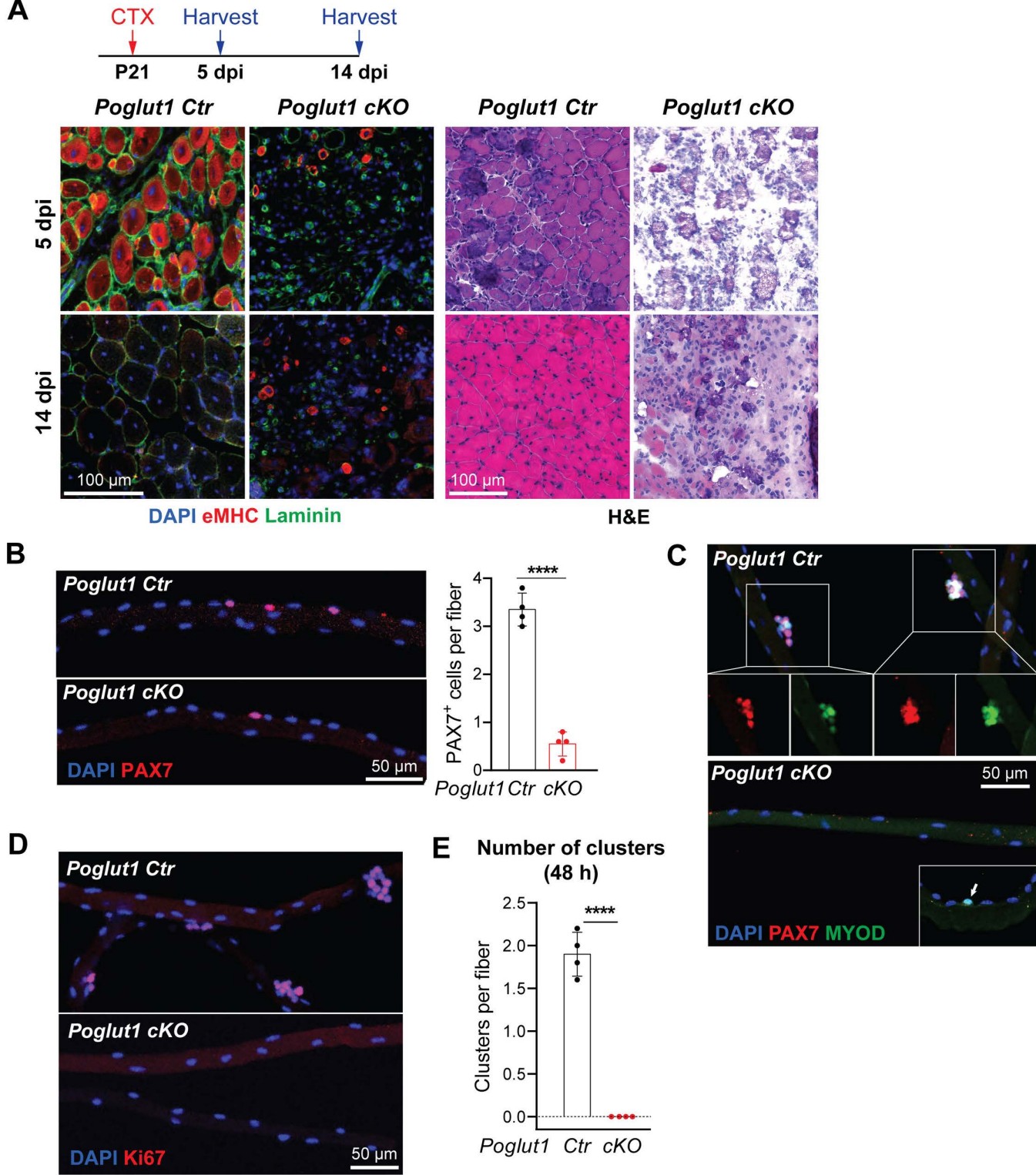

**Fig 4. *Poglut1*-deficient muscles do not exhibit proper satellite cell activation and muscle repair upon muscle injury. A)** P21 mouse TA muscles were injected with CTX and harvested either 5 or 14 days post injection. Immunofluorescent staining [DAPI (blue), Laminin (green), and eMHC (red)] and H&E images of injured and non-injured TA muscles of control and mutant animals are shown. **B)** Staining of isolated myofibers from P21 mice with DAPI

(blue) and PAX7 (red) and quantification of PAX7⁺ cells per myofiber. **C)** Staining of isolated fibers after 48 hours of culture for DAPI (blue), PAX7 (red), and MYOD (green). The arrow in the inset of the cKO panel marks a MYOD⁺ single cell occasionally seen in the cKO myofibers at this time. **D)** Staining of isolated fibers after 48 hours of culture for DAPI (blue) and Ki67(red). **E)** Quantification of clusters present in myofibers cultured for 48 hours post isolation. Siblings without *Poglut1* deletion (*Poglut1⁺/⁺*, *Poglut1ᶠˡᵒˣ/ᶠˡᵒˣ*, or *Pax7ᶜʳᵉ; Poglut1⁺/⁺*) were used as controls in all experiments. In B and E, each dot represents an animal, mean±SD, unpaired *t* test. ****$P < 0.0001$.

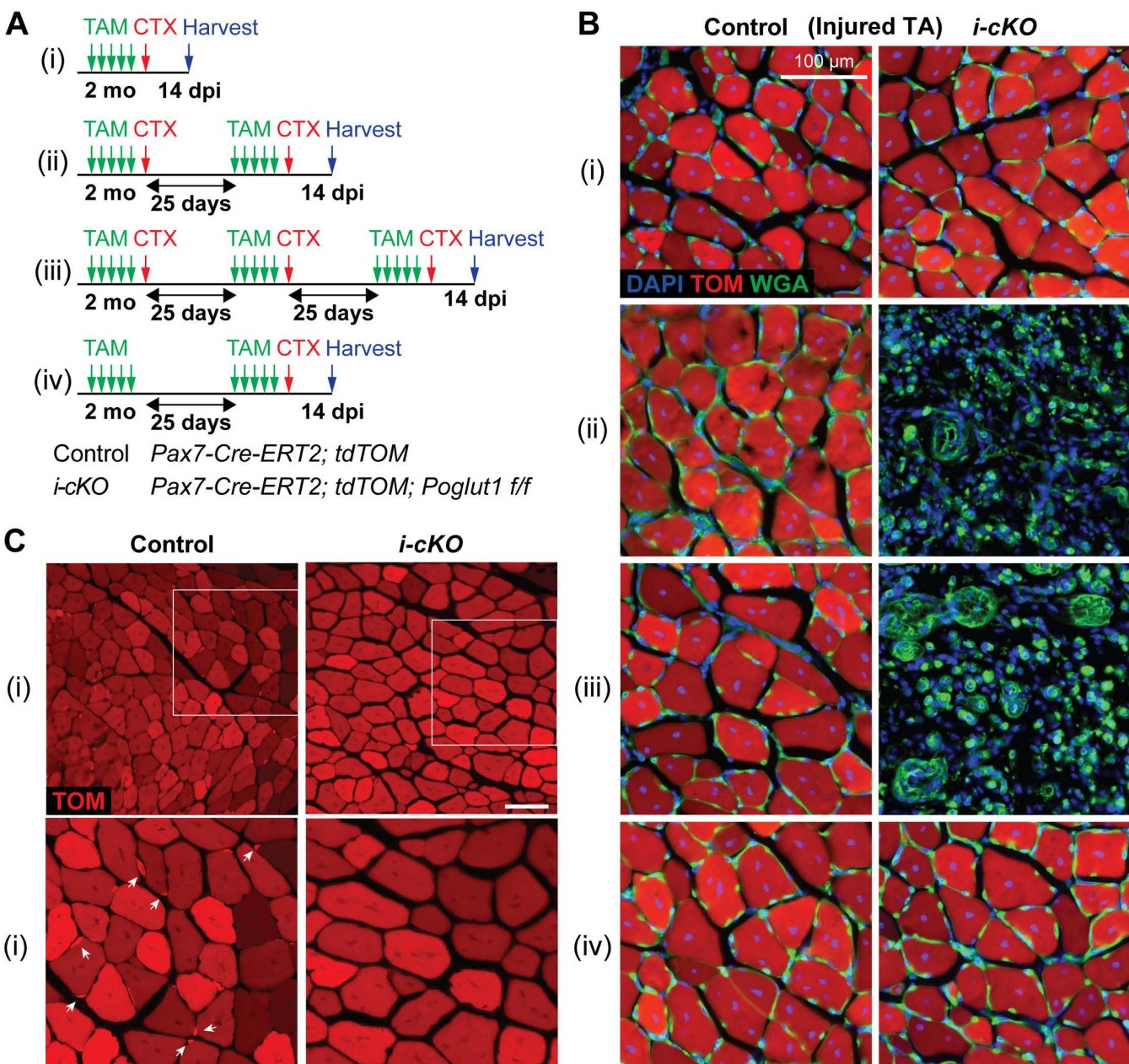

**Fig 5. *Poglut1* is required for adult satellite cell self-renewal after cardiotoxin injury. A)** Schematics of the tamoxifen (TAM) and cardiotoxin (CTX) injection regimens in TA muscles of two-month-old mice. **B)** Sections showing tdTomato (TOM, red) expression with WGA (green) and DAPI (blue) after single or multiple CTX injuries as shown in **A. C)** TOM signal in TA muscles of *i-cKO* and control mice after one round of injury (i in **A)**. Arrows in the Control closeup mark single TOM⁺ cells at the periphery of repaired myofibers. n=4 per condition.

However, *i-cKO* muscles did not show any repair in response to the second and third round of injury, while the control mice showed full repair in both cases (Fig 5Bii and 5Biii). These observations suggested to us that loss of *Poglut1* might impair the self-renewal of PAX7$^+$ satellite cells. To further test this notion, we performed two additional experiments. First, we performed two rounds of tamoxifen injections in a new cohort but skipped the first round of injury and only induced CTX injury after the second round of recombination (Fig 5Aiv). Analysis of 14 dpi muscle showed a full repair in both control and *i-cKO* mice (Fig 5Biv). Second, we inspected the repaired control and *i-cKO* muscle after one round of recombination and injury (Fig 5Ai) for the presence of single tdTomato$^+$ cells next to the repaired myofibers, which would represent the satellite cells formed after the repair. While single tdTomato$^+$ cells were readily seen in the repaired muscle in control mice, we did not see any such cells next to the repaired muscle from *i-cKO* animals (Fig 5C, n = 4 animals per condition). Together, these observations provided strong evidence that loss of *Poglut1* in adult muscle stem cells impairs their ability to return to quiescence and form new satellite cells upon repair of injured muscle.

To assess the role of *Poglut1* in satellite cell maintenance, we analyzed the expression pattern of tdTomato in uninjured TA muscles of *Poglut1-i-cKO* and control animals. We observed two types of tdTomato signal in these samples: small tdTomato$^+$ cells, which based on their location are satellite cells (Fig 6A, arrowheads); and tdTomato$^+$ myofibers (Fig 6A), which indicate fusion of tdTomato$^+$ satellite cell(s) with the corresponding myofiber. In control muscles, the number of small tdTomato$^+$ cells was reduced in the 2xTAM cohort compared to the 1XTAM cohort, without further decline between 2XTAM and 3XTAM (Fig 6B). This suggests that the number of satellite cells in mouse TA muscles reaches a plateau between 3–4 months of age. Importantly, in all three cohorts the number of small tdTomato$^+$ cells in *Poglut1-i-cKO* muscles was significantly less than that in control muscles (Fig 6B). Both *i-cKO* and control muscles harbored tdTomato$^+$ myofibers in all three cohorts. However, in 2XTAM and 3XTAM cohorts, the number of tdTomato$^+$ myofibers with strong tdTomato signal intensity was significantly higher in *i-cKO* muscles compared to control muscles from the same cohort, at the expense of myofibers with minimal to no tdTomato expression (Fig 6C). Together, these observations indicate that loss of *Poglut1* in adult satellite cells leads to their loss over time due to accelerated differentiation and fusion.

We next examined the impact of loss of *Poglut1* in adult satellite cells on their quiescence. PAX7 staining of uninjured TA muscles after one round of tamoxifen injection indicated that in *i-cKO* and control animals, 93% and 97% of tdTomato$^+$ cells expressed PAX7, respectively (Fig 6D and 6E). Using tdTomato as a surrogate for PAX7, we then co-stained the samples with MYOD and Ki67. *i-cKO* muscles showed a strong and statistically significant reduction in the percentage of tdTomato$^+$ cells that are MYOD$^-$ Ki67$^-$ and therefore fit the description of quiescent satellite cells (Fig 6F and 6G). This was accompanied by a significant increase in the percentage of tdTomato$^+$ cycling muscle stem cells (MYOD$^-$ Ki67$^+$) and non-cycling muscle progenitors (MYOD$^+$ Ki67$^-$) in *i-cKO* TA muscles (Fig 6G). These data strongly suggest that deletion of *Poglut1* leads to loss of quiescence in adult satellite cells.

### *Poglut1* knockdown leads to reduced NOTCH1, 2, and 3 signaling in cell-based signaling assays in C2C12 cells

We and others have previously shown that *Poglut1* knockdown leads to reduced NOTCH1 signaling in several mammalian cell lines, including C2C12 myoblasts [16,33–35]. In agreement with these observations, satellite cells freshly isolated from *Poglut1-cKO* muscles and directly used in immunoblot assays showed a strong reduction in the level of cleaved (active) NOTCH1 intracellular domain compared to control satellite cells (Fig 7A). Moreover, we have recently reported that loss of one copy of *Poglut1* in the signal-receiving cells leads to a mild but statistically significant reduction in JAG1-induced NOTCH2 signaling in co-culture assays [36]. However, we also reported that reducing *Poglut1* in the signal-sending cells can increase the JAG1 protein level and activation of NOTCH1 and NOTCH2 in neighboring cells [36]. Moreover, the role of POGLUT1 in NOTCH3 signaling was not known and it was not known whether DLL1-mediated NOTCH2 signaling also depended on the expression of *Poglut1*. Accordingly, we sought to systematically examine the effects of reducing POGLUT1 in the signal-receiving cells overexpressing mouse NOTCH1, NOTCH2, or NOTCH3 in response to DLL1 and JAG1 ligands. In these assays, we used C2C12 myoblasts stably transfected with an shRNA

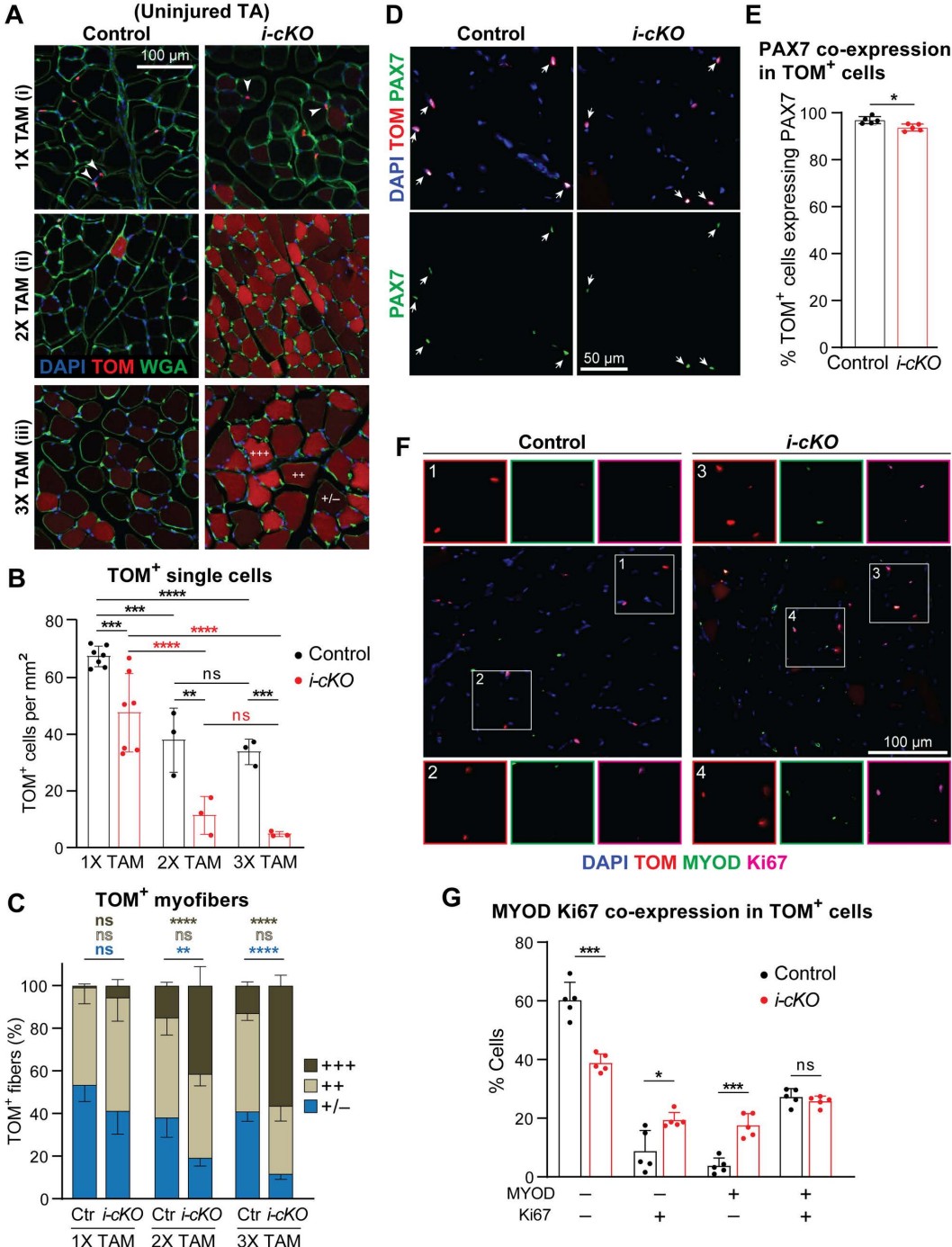

**Fig 6. *Poglut1* is required for the maintenance of quiescent adult satellite cells. A)** Sections of uninjured TA muscles showing TOM (red), WGA (green) and DAPI (blue) after single or multiple TAM inductions as described in Fig 5A. TOM⁺ single cells are indicated by arrowheads. Intensity of TOM signal in myofibers is scored as follows: (+++) strong, (++) moderate, and (+/–) minimal to no expression. **B)** Quantification of TOM⁺ single cells per mm² of uninjured TA sections after single and multiple rounds of TAM induction. **C)** Quantification of percentage of TOM⁺ myofibers with various TOM intensities after single and multiple TAM inductions. The graph shows strong (dark brown), moderate (light brown), and minimal to no (blue) expression, with corresponding statistics in the same colors labeled on top. **D)** Immunostaining for PAX7 (green) in uninjured TA sections of control and *i-cKO* animals after one round of TAM induction. Arrows point to cells double-positive for PAX7 and TOM (red). **E)** Quantification of the percentage of TOM⁺ cells co-expressing PAX7 in control and *i-cKO* TA sections after one round of TAM. **F)** Immunostaining for MYOD (green) and Ki67 (magenta) in uninjured TA

sections of control and *i-cKO* animals after one round of TAM. **G)** Quantification of the percentage of TOM[+] cells expressing MYOD, Ki67, both or none. Each dot represents an animal in B, E, **G**. Mean±SD; two-way ANOVA with Šidák's (B) or Tukey's (C) multiple comparisons test and unpaired *t* test **(E, G)**. ns: not significant, *$P < 0.05$, ***$P < 0.001$ and ****$P < 0.0001$.

targeting *Poglut1* and control. C2C12 cells stably transfected with an shRNA sequence that does not target any mouse genes (non-target; NT). In agreement with our original report on these cells [16], C2C12 cells with shRNA-*Poglut1* showed a strong reduction in *Poglut1* expression compared to control cells (Fig 7B). Importantly, the shRNA-*Poglut1* cells also exhibited a 61% reduction in *Pax7* expression, further supporting their usage to study the impact of reducing *Poglut1* on Notch signaling in myoblasts. The cells were co-transfected with a luciferase-based Notch reporter plasmid and a β-galactosidase expression plasmid as transfection efficiency control and cultured on plates coated with DLL1-IgG1-Fc, JAG1-IgG1-Fc, or IgG1-Fc alone as a negative control. As shown in Fig 7C, *Poglut1* knockdown cells showed a significant reduction in signaling mediated by all three Notch receptors examined here in response to both JAG1 and DLL1. We conclude that POGLUT1 in the signal-receiving cell promotes the activation of all three Notch receptors involved in myogenesis.

### Mouse NOTCH3 is glycosylated by POGLUT1

We have previously reported efficient glycosylation of mouse NOTCH1 and mouse NOTCH2 by POGLUT1 [13,16]. Mouse NOTCH3 has 15 predicted POGLUT1 target sites [16]. However, although several EGF repeats of human NOTCH3 were shown to be *O*-glucosylated by POGLUT1 [37], it is not known whether mouse NOTCH3 is modified by this enzyme, and a comprehensive analysis of mammalian NOTCH3 modification by POGLUT1 has not been reported. Given the strong impact that *Poglut1* knockdown showed on NOTCH3 signaling (Fig 7C), we performed mass spectrometric analysis on mouse NOTCH3 expressed in HEK293T cells. We were able to map 13 of the mouse NOTCH3 EGF repeats with a consensus POGLUT1 modification site and observed *O*-linked glucosylation at high stoichiometry on all of them (Figs 7D and S5). Similar to our previous reports on other POGLUT1 targets [18,38], some mouse NOTCH3 EGF repeats primarily harbored a fully extended xylose-xylose-glucose-*O* trisaccharide, while others had *O*-glucose monosaccharide only or a combination of mono-, di- and trisaccharide (S5 Fig). These observations indicate that POGLUT1 efficiently glycosylates mouse NOTCH3 and that the degree of extension to di- and trisaccharide is different in different EGF repeats.

### Discussion

Recessive variants in human *POGLUT1* were previously shown to cause LGMDR21, which is associated with a severe reduction of PAX7[+] satellite cells and reduced NOTCH1 signaling in patient muscles [10,11]. While the disease is adult-onset in most patients, some cases start at a very young age [10,11]. Moreover, strong *Poglut1* knockdown (81%) in C2C12 myoblasts was reported to lead to premature differentiation and enhanced fusion of these cells [35]. These observations, combined with our recent report on myogenic progenitors derived from LGMDR21 patient iPSCs [14], suggested that POGLUT1 might function in satellite cells to promote muscle maintenance and potentially muscle development. However, *in vivo* evidence in a mammalian model organism for the role of POGLUT1 in the developing myogenic progenitors and in adult satellite cells was lacking.

To answer these questions, we performed conditional knockout and inducible conditional knockout studies to delete *Poglut1* from PAX7[+] cells during muscle development and in adult mice, respectively. Our data indicate that in both contexts, loss of *Poglut1* leads to premature differentiation of the stem/progenitor cells and their fusion with myofibers. Moreover, *ex vivo* culture of myoblasts and EDL myofibers indicates a significant reduction in the proliferative capacity of muscle progenitors upon loss of *Poglut1*. We propose that these abnormalities lead to a severe depletion of the stem/progenitor pool. In early postnatal mice, this rapid reduction in the number of PAX7[+] cells deprives the growing muscles from

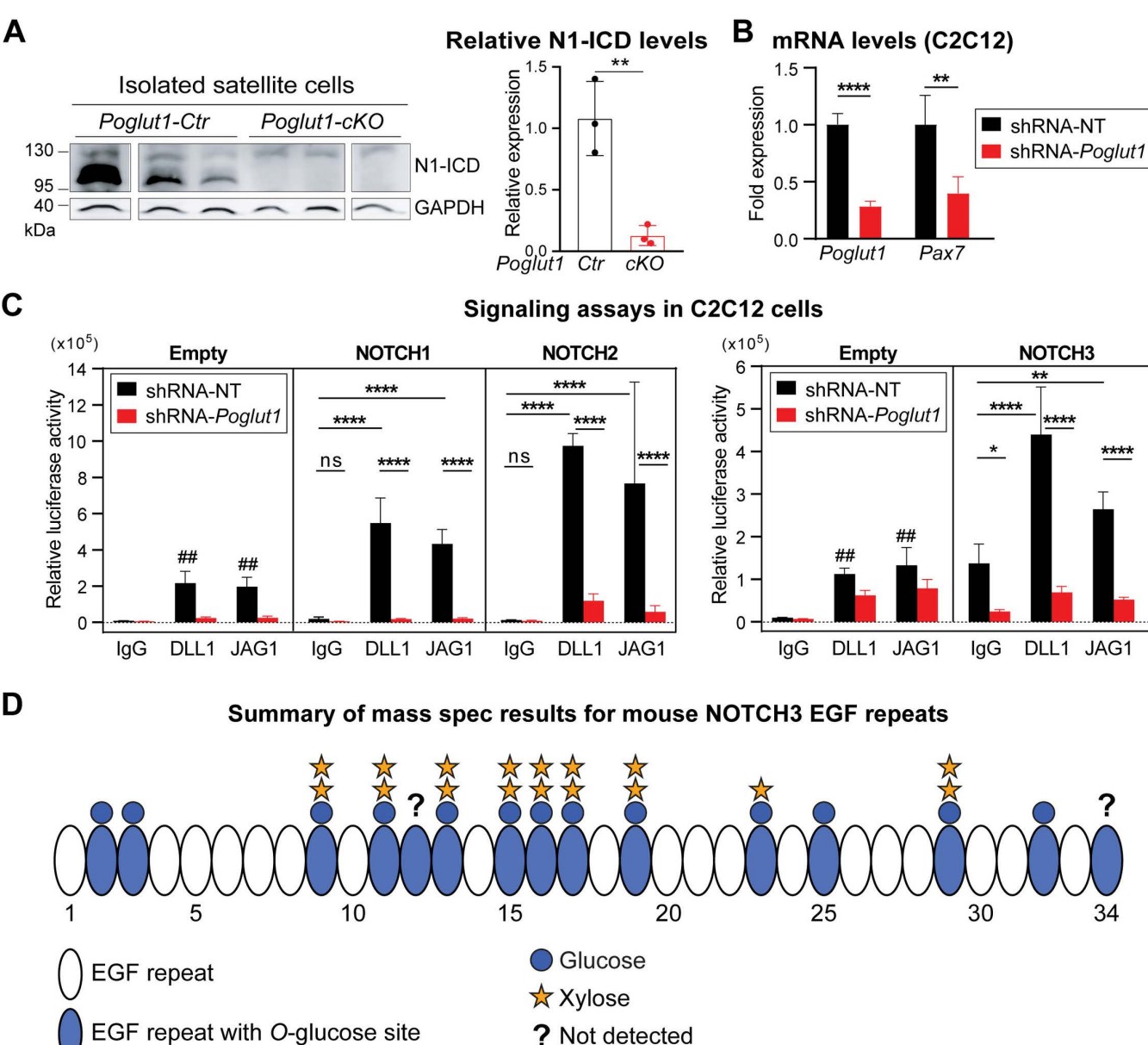

**Fig 7. *Poglut1* is required for JAG1- and DLL1-mediated signaling through NOTCH1, 2 and 3, and NOTCH3 is a POGLUT1 target protein. A)** Western blot image using anti-V1744 antibody shows the expression level of cleaved (active) NOTCH1 intracellular domain (N1-ICD) in satellite cells isolated from muscles of control versus *Poglut1-cKO* mice with GAPDH as loading control. **B)** *Poglut1* and *Pax7* mRNA levels measured by qRT-PCR in C2C12 cells stably expressing an shRNA against *Poglut1* or a non-target (NT) control shRNA. Unpaired *t* test. **\*\*P < 0.01, \*\*\*\*P < 0.0001. C)** Cell-based signaling assays using the above-mentioned C2C12 cells cultured on plates coated with JAG1, DLL1 or IgG as control to induce signaling from endogenous (empty vector) or overexpressed Notch receptors. Relative luciferase expression levels normalized for transfection efficiency are shown. Three-way ANOVA with Tukey's multiple comparisons test. Each circle represents a cell culture well. Each graph is representative of at least 2 independent experiments. Note that for each dataset, DLL1- and JAG1-induced signaling upon NOTCH receptor overexpression was significantly higher than that induced by the same ligand in the control (Empty vector) samples (##). Mean±SD, ns: not significant, \*P < 0.05, \*\*P < 0.01, and \*\*\*\*P < 0.0001. **D)** Schematic of the EGF repeats of mouse NOTCH3 decorated with glycan structures based on mass spectrometry data. For each EGF repeat, the most prevalent glycoform is shown. Mass spectrometry traces and glycoform quantifications are shown in S5 Fig. EGF repeats labeled with a question mark were not detected in mass spectrometry data.

maintaining sufficient muscle progenitors, which are required for proper muscle growth and for generation of an adequate number of satellite cells [27]. Moreover, loss of *Poglut1* in PAX7+ cells negatively impacted the ability of the tibialis anterior muscle to repair cardiotoxin-induced damage in both contexts. Together, these observations establish that expression of *Poglut1* in PAX7+ cells is essential for normal development and maintenance of satellite cells and for muscle repair.

We had previously shown that POGLUT1 glycosylates multiple EGF repeats harboring the CXSXPC consensus sequence in a number of insect and mammalian proteins, including *Drosophila* Notch and mammalian NOTCH1 and NOTCH2 [12,13,16–18,38]. In addition, a recent report indicated that several EGF repeats in human NOTCH3 are *O*-glucosylated by POGLUT1 [37]. Moreover, *O*-glucosylation of *Drosophila* Notch and mammalian NOTCH1 and NOTCH2 was previously shown to promote the activation of these receptors (only in response to JAG1 in the case of NOTCH2) [12,13,36,39]. The current work extends these data by showing that expression of POGLUT1 in the signal-receiving cells promotes both JAG1- and DLL1-mediated signaling by NOTCH1, NOTCH2, and NOTCH3 receptors, all of which play key roles in muscle stem/progenitor cells [3]. Furthermore, our data indicate that multiple EGF repeats of the mouse NOTCH3 are also *O*-glucosylated at their POGLUT1 consensus site, strongly suggesting that *O*-glucosylation of NOTCH3 by POGLUT1 promotes its ligand-mediated activation. Moreover, loss of *Poglut1* with *Pax7-Cre* and *Poglut1* knockdown in C2C12 myoblasts both lead to a severe reduction in expression of *Pax7*, which was previously shown to be a direct target of Notch signaling in satellite cells [40]. These data, along with the similarities between the muscle phenotypes of *Poglut1-cKO* and *i-cKO* (this study) and the phenotypes observed in *Pax7*−/− animals [41,42] and animals with germline or conditional loss of various Notch pathway components [4–9] strongly suggest that reduced *Pax7* expression due to reduced Notch signaling contributes to the muscle phenotypes caused by loss of *Poglut1* in muscle stem/progenitor cells.

At the late prenatal and early postnatal stages, *Poglut1-cKO* animals showed abnormalities in the extracellular matrix proteins laminin and collagen IV, both of which are thought to directly contact the satellite cells as part of their niche [2]. These observations suggest that the premature differentiation of *Poglut1-cKO* muscle progenitors is at least in part due to abnormalities in their niche, a phenomenon which has been previously linked to impaired Notch signaling in the developing muscle [43]. This notion is also in agreement with our recent RNA-sequencing data, which identified ECM components to be among the top differentially expressed genes in LGMDR21 myogenic cells [14]. Additionally, our data strongly suggest that upon loss of *Poglut1*, adult satellite cells fail to remain in quiescence and are spontaneously activated. Notch signaling has been shown to directly activate the transcription of several genes encoding collagen V and VI in muscle stem/progenitor cells, and collagen V expressed by satellite cells plays a key role in the niche to maintain satellite cell quiescence [44]. Therefore, ECM abnormalities caused by reduced Notch signaling might contribute to the loss of satellite cell quiescence in *i-cKO* animals as well. Of note, the first round of injury can be repaired in the *i-cKO* animals even after two rounds of tamoxifen-induced *Poglut1* deletion in satellite cells, but the subsequent injuries cannot be repaired. Moreover, despite muscle repair, small tdTomato+ cells likely to be the newly formed satellite cells cannot be found in *i-cKO* TA muscles. These observations suggest that despite loss of *Poglut1*, satellite cells can be activated upon injury and repair the damage but fail to self-renew due to reduced Notch signaling, leading to a failure in muscle repair upon subsequent injury.

In summary, our data suggest that expression of the glycosyltransferase POGLUT1 in myogenic progenitors promotes their proliferation and prevents their precocious differentiation, thereby playing an essential role for postnatal muscle growth and establishment of muscle stem cells capable of repairing muscle after injury. Moreover, POGLUT1 is essential for the maintenance of adult satellite cells and their self-renewal capacity after activation upon muscle injury. Mechanistically, our data suggest that a reduction in the activity of all three Notch receptors involved in satellite cell biology and the resulting decrease in the expression of PAX7 contribute to *Poglut1* loss-of-function phenotypes, although the impact of other POGLUT1 targets in these phenotypes need to be examined. These observations provide novel insight into the role of glycosylation in myogenesis and suggest a potential mechanism underlying the loss of satellite cells in LGMDR21 patients.

## Materials and methods

### Ethics statement

Experiments and procedures with animals were designed to minimize the animal suffering and reduce the number of animals used. All experiments were performed in accordance with the Spanish and the European Union regulations (RD53/2013 and 2010/63/UE) and approved by Animal Experimentation Ethics Committee (CEEA)/Research Ethics Committee of the Hospitals Virgen Macarena - Virgen del Rocío, as well as with the guidelines of Animal Care and Use Committee of Baylor College of Medicine under approved animal protocols.

### Mouse strains, breeding, and genotyping

Animals were housed under controlled temperature and humidity conditions, alternating 12-hour light cycles, at barrier facilities at Baylor College of Medicine and animal facilities at Hospital Virgen Macarena/University of Seville.

The following strains were used: wild-type C57BL/6, *Poglut1*$^{flox}$ ( [18]; official name *Poglut1*$^{tm1.1Hjnd}$), *Pax7*$^{Cre}$ ( [15]; JAX #010530; official name *Pax7*$^{tm1(cre)Mrc}$/J), *Pax7*$^{CreERT2}$ ( [31]; JAX #017763; official name B6.Cg-*Pax7*$^{tm1(cre/ERT2)Gaka}$/J), and *ROSA-CAG-lox-stop-lox-tdTomato*-Ai9 ([32]; JAX #007909; official name B6.Cg-*Gt(ROSA)26Sor*$^{tm9(CAG-tdTomato)Hze}$/J). All animals were kept on a C57BL/6 background. To generate *Poglut1-cKO* animals, crosses were set between *Pax-7*$^{Cre}$; *Poglut1*$^{+/flox}$ and *Poglut1*$^{flox/flox}$ or *Poglut1*$^{+/flox}$ animals. Siblings without *Poglut1* deletion (*Poglut1*$^{+/+}$, *Poglut1*$^{flox/flox}$, or *Pax7*$^{Cre}$; *Poglut1*$^{+/+}$) were used as control in all experiments involving *Poglut1-cKO* mice. To generate *Poglut1-i-cKO* animals, *Pax7*$^{CreERT2}$; *Poglut1*$^{flox/flox}$ and *ROSA-CAG-lox-stop-lox-tdTomato*-Ai9; *Poglut1*$^{flox/flox}$ animals were crossed to each other. Two-month-old *Pax7*$^{CreERT2}$; *Poglut1*$^{flox/flox}$; *ROSA-CAG-lox-stop-lox-tdTomato*-Ai9 progeny from this cross received five daily intraperitoneal injections of tamoxifen for one, two, or three rounds, with the resulting animals called *Poglut1-i-cKO*. As control, we used *Pax7*$^{CreERT2}$; *ROSA-CAG-lox-stop-lox-tdTomato*-Ai9 animals from a parallel cross injected with the same tamoxifen regimen. Genomic DNA from tail tips was used for PCR genotyping.

### Behavioral assays

Open Field Test. The locomotor activity was measured every other day starting at P9, by placing each mouse individually in the center of a square box ($45 \times 45\,cm^2$) with 45-cm high walls and allowing the animal to explore for 5 minutes. The mouse activity was digitally recorded using a video camera placed above the center of the arena. The box was thoroughly cleaned with 70% ethanol between trials. The Record-it Media v1.0 software was used to track the mouse activity. The center of mass of each mouse was established as the detection point for the software to recognize the mouse. The data was analyzed using SMART v3.0 software. We collected data from general locomotion and quantified the maximum speed (cm/s). We set 1.10 cm/sec as slow moving threshold and 14.20 cm/sec as fast moving threshold, in order to apply the same analyses parameters to all the analyzed animals.

Body Weight. All mice were weighed on alternating days, using an electronic scale (GRAM EH-1000). The data was collected at the same time each day and always before subjecting the animal to further behavioral tests. Body weight was measured starting at P7.

Grip test. The grip strength test was carried out every other day in mice after the neonatal stage. We start at P12 until P31, using a grip test instrument (Bioseb). To perform this test, the mouse was positioned over the grid horizontally held by the base of the tail and was allowed to grasp onto the grid with both front limbs. Once the animal was attached and stable, we pulled back in parallel to the grid, analyzing only the strength of the front limbs. The grid was thoroughly cleaned with 70% ethanol after each trial. Only the highest strength value obtained during each test was used for statistical analysis. The strength was measured in grams (2500 g = 25 N). This procedure was repeated three times to obtain consistent data.

## Muscle electrophysiology

Mice were euthanized with $CO_2$ and exsanguinated. The levator auris longus (LAL) muscle was dissected with its nerve branches intact and pinned in a 2 mL chamber lined with silicone rubber. Preparations were continuously perfused with a solution of the following composition: 125 mM NaCl, 5 mM KCl, 2 mM $CaCl_2$, 1 mM $MgCl_2$, 25 mM $NaHCO_3$ and 15 mM glucose. The solution was continuously gassed with 95% $O_2$ and 5% $CO_2$.

Intracellular recordings of end plate potentials (EPPs) were performed at the LAL muscle as previously described [45]. Muscle contractions were prevented by including in the bath 3–4 µM m-conotoxin GIIIB (Alomone Laboratories). The mean amplitudes of the EPP and miniature EPPs (mEPPs) recorded at each NMJ were linearly normalized to -70 mV resting membrane potential and EPPs corrected for nonlinear summation [46,47]. Quantal content (QC) was estimated by the direct method, which consists of recording mEPPs and EPPs (nerve stimulation 0.5 Hz) simultaneously and then calculating the ratio: QC = Average Peak EPP/ Average Peak mEPP. All data are given as groups mean values ± SD. All experiments reported were done at room temperature and include the results of at least three animals per genotype.

## Mouse muscle immunofluorescent staining and quantifications

After muscle extraction from hind limbs at the indicated ages and time points, TA were fixed at 4 °C with 4% paraformaldehyde for 24 hours, and cryoprotected with 30% sucrose before undergoing freezing at 80 °C. Ten-µm thick sections were placed serially onto Superfrost Plus microscope slides. For antigen retrieval, sections were incubated for 6 minutes in pH 6.6 citrate buffer at approximately 100 °C. Once cooled, sections were permeabilized with 0.2% Triton X-100 for 15 minutes and blocked with 2% BSA and Mouse on Mouse Blocking Reagent (Vector Laboratories, MKB-2213) for 1 hour. The following antibodies were used: mouse monoclonal anti-PAX7 (DSHB, PAX7, 1:2.5), rabbit anti-laminin (Sigma, L9393, 1:200), mouse anti-laminin (LAM-89, Sigma, 1:100), rabbit anti-collagen VI (ab6588, Abcam, 1:500), rabbit anti-M-cadherin (Cell Signaling, 40491, 1:500), mouse anti-eMHC (DSHB, F1.652, 1:2.5), mouse anti-MYOD (DSHB, D7F2, 1:2.5), rabbit anti-MYOD (Invitrogen, PA5–23078, 1:200), rabbit anti-Ki67 (Sigma, AB9260, 1:1000), goat anti-mouse IgG (H + L) Highly Cross Absorbed Secondary Antibody Alexa Fluor 555 (Invitrogen, A-21424, 1:500), and Goat anti-rabbit IgG (H + L) Highly Cross Absorbed Secondary Antibody Alexa Fluor 488 (Invitrogen, A-11008, 1:500). Samples were also labeled with wheat germ agglutinin (WGA) CF488A (Biotium, 29022, 1:1,000), Phalloidin Alexa Fluor 488 (Invitrogen, A12379, 1:1000) and/or DAPI (Invitrogen, D1306). To detect the tdTomato signal, fresh sections of muscle samples were incubated with WGA and DAPI without antigen retrieval process. Fluorescent images were obtained using Leica DFC365 FX or Leica Stellaris 8 and processed with the ImageJ software.

For quantification of cells, the number of cells were manually counted from at least 5 images for each animal. Those numbers were averaged and then scaled to match the area of 1 mm². This was consistently performed for quantifying PAX7+ cells, M-Cad+ cells, and eMHC+ fibers. The percentage of myofibers with internal nuclei was determined by dividing the number of myofibers with internal nuclei by the total number of myofibers in each image. Image analysis for measuring myofiber CSA was performed using the MuscleJ plugin for ImageJ [48].

## Cardiotoxin injury

Cardiotoxin (CTX) (Latoxan, L8102) was prepared by diluting in sterile PBS to a final concentration of 10 µM and injected into left tibialis anterior (TA) of mice using micro fine insulin syringes (30G x 8mm). Mice were maintained anesthetized using 2% Isoflurane during the whole procedure (5–10 min). For experiments shown in Fig 4A, CTX was injected at P21, and animals were sacrificed at 5 and 14 days post-injection. For experiments shown in Fig 5A–5B, CTX injections started at 2 months of age, followed by 1 or 2 cycles of re-injections (refer to Fig 5 for specific timeline). The injured and uninjured TA were harvested, fixed in 4% paraformaldehyde for 24 hours at 4 °C and cryoprotected in sucrose, before snap-freezing in liquid nitrogen and storing at −80 °C. The TA were cut transversally into 10-µm sections using a Cryostat (Leica) and placed onto Superfrost Plus microscope slides.

## Hematoxylin and Eosin staining

TA sections were allowed to reach room temperature, and immediately fixed with 4% paraformaldehyde for 20 minutes at room temperature. After washing twice with PBS, slices were incubated in hematoxylin solution for 3 minutes, followed by incubation with lithium carbonate for 2 minutes, all at room temperature. To eliminate residues, samples were rinsed under indirect running water for 4 minutes. Hereafter, muscle slices were incubated in eosin for 2–3 minutes and rinsed twice with $_{dd}H_2O$. Before mounting, tissue was dehydrated by running the slides through increasing concentrations of alcohol. Briefly: 2 min in Ethanol 70%, 2 min Ethanol 96%, 2 min Ethanol 96%, 5 min Ethanol 100%, 5 min Xylenol, 5 min Xylenol. Finally, the slices were air dried and mounted using DPX (06522, Sigma-Aldrich).

## Single myofiber isolation, culture, staining, and quantifications

Myofiber isolation was performed as described previously [49]. Briefly, the TA muscle of mice was first removed to expose the extensor digitorum longus (EDL) muscle which was carefully isolated and digested in collagenase (400 U/mL) at 37°C for 1 hour. The EDL muscle was gently triturated to dissociate single myofibers. The fibers were left in the collagenase solution until most single myofibers were separated. Once most fibers were detached from each other, they were either fixed for immunostaining or transferred to media for 48 hours of culture. These myofibers were fixed in 4% PFA for 10 minutes, washed 3 times with 1X PBS, then permeabilized with 0.2% PBST for 10 minutes. They were then blocked with 5% Goat serum for 1 hour in the dark, followed by primary antibody staining overnight at 4 °C. For each animal, 5–6 myofibers were used for quantification.

## Primary satellite cell isolation and culture

Isolation of satellite cells from *Poglut1-cKO* mouse skeletal muscle was performed as described by the manufacturer with minor modifications (#130-098-305, #130-104-268; Miltenyi Biotec). Briefly, forelimb and hindlimb muscles from P4 *Poglut1-cKO* and sibling controls were dissociated using the mouse Skeletal Muscle Dissociation Kit and the gentle-MACS Dissociator (Miltenyi Biotec, Bergisch Gladbach, Germany). Subsequently, the homogenate was incubated with microbeads conjugated to a cocktail of monoclonal antibodies against no-target cells (Miltenyi Biotec, Bergisch Gladbach, Germany) at 2–8 °C for 20 min. The cell suspension was applied onto the MACS Column placed in a MACS Separator (Miltenyi Biotec, Bergisch Gladbach, Germany). We first collected the flow-through containing unlabeled cells, representing the enriched satellite cells fraction. We also collected the labelled cell fraction, as control for this experiment. Satellite cells, once cultured in proliferation medium containing 66% DMEM, 20% fetal bovine serum (FBS), 10% horse serum, 1 mM sodium pyruvate, 1 mM Hepes; 2 mM glutamine, 100 U/mL penicillin/ 100 U/mL streptomycin, 2.5 ng/mL mouse fibroblast growth factor basic (bFGF), proliferated and differentiated to myoblast (myogenic progenitor cells). Myoblasts were then seeded at 8,000 cells/cm$^2$ using the proliferation medium for the myoblast stage. We examined proliferation at different time points (1, 3 and 5 days). When the myoblast cultures reached confluency, the medium was substituted by differentiation medium (95% DMEM, 5% horse serum, 100 U/mL penicillin/ 100 U/mL streptomycin). The myofusion index was determined three days later by calculating the mean percentage of nuclei in myotubes relative to the total number of nuclei (myoblasts + myotubes). The following primary antibodies were used: mouse monoclonal anti-PAX7 (DHSB, 1:25); rabbit polyclonal anti-MYOD (LifeSpan Biosciences, LS-B9421,1:50); rabbit monoclonal anti-desmin (Abcam, D93FD5, 1:100); mouse monoclonal anti-Myogenin (Abcam, F5D, 1:100); and rat monoclonal anti-Ki67 (Invitrogen, 740008T, 1:200).

## RNA isolation and Real Time qRT-PCR

Frozen muscle tissue and isolated satellite cells from cKO and controls were used for total RNA isolation using TRIzol Reagent (Invitrogen) following manufacturer specifications. Once isolated, RNA was purified using E.Z.N.A HP Total RNA Isolation kit (Omega) and quantified with a NanoDrop 2000 spectrophotometer (Thermo Fischer). Reverse transcription

was performed using PrimeScript RT Master Mix (Takara) following the recommended protocol and using equal amounts of RNA in all samples (500 ng RNA/10-µL reaction).

Real time RT-PCR was performed using equal volumes of cDNA samples taken directly from the reverse transcription reaction (0.1 µL cDNA per RT-PCR reaction). For the preparation of the mix we used TaqMan Fast Advanced Master Mix combined with TaqMan Gene Expression Assay Probe (Applied Biosystems). Amplification and measurement were performed by ABI PRISM Sequence Detection System 7900 (Applied Biosystems). Ct values were calculated by the software provided by Applied Biosystems (SDS 2.0). The reaction was first incubated at 50 °C for 2 min, 95 °C for 10 min, and then 40 cycles at 95 °C for 15 s and 60 °C for 1 min. For the quantification of cDNA level, we used the comparative double delta cycle threshold (Ct) method ($2^{-\Delta\Delta Ct}$). We determined *Pax7* (Mm01354484_m1) expression using *Gapdh* (Mm99999915_g1) as a housekeeping gene to normalize the cDNA expression levels.

### C2C12 cell culture and Notch signaling assays

To activate the Notch receptors, cell culture plates were pre-coated with recombinant human DLL1-IgG1-Fc (Biotechne R&D systems, 10184-DL), human JAG1-IgG1-Fc (Biotechne R&D systems, 1277-JG), or human IgG1-Fc (Biotechne R&D systems, 110-HG) as negative control. Recombinant proteins were diluted in sterile PBS at 2 µg/mL for DLL1-IgG1-Fc, and 4 µg/mL for JAG1-IgG1-Fc and IgG1-Fc control and incubated in 24-well culture plates for 2 hours at room temperature. C2C12 cells stably transfected with an shRNA targeting *Poglut1* and control C2C12 cells stably transfected with a non-target shRNA (NT) were described previously [16]. After gently washing with PBS, the cells were plated on ligand-coated or IgG1-coated plates and transfected with the TP-1 luciferase Notch-signaling reporter construct (0.12 µg/well), plasmids expressing WT mouse NOTCH1, NOTCH2, NOTCH3 or empty pcDNA3 (0.1 µg/well), and gWIZ β-galactosidase construct (0.06 µg/well). The latter was used for transfection efficiency normalization using lipofectamine 2000 (Invitrogen) according to the manufacturer's instructions. After 48 hours of transfection, cells were lysed and luciferase and β-galactosidase assays (Luciferase Assay System; Promega) were performed according to the manufacturer's instructions. Given the variability in luciferase-based co-culture assays, 2–3 independent experiments were performed for each ligand-receptor pair, with each experiment comprised of four replicates (wells). The following probes were used to quantify the relative mRNA expression in C2C12 cells: *Poglut1* (Mm00552419_m1) and *Pax7* (Mm01354484_m1).

### Western blotting and laminin overlay assays

Isolated satellite cells and frozen muscle samples were homogenized in RIPA buffer (20 mM Tris–HCl pH 7.4, 150 mM NaCl, 1 mM EDTA, 1% IGEPAL, 0.1% SDS) containing protease inhibitor mixture (Roche). The lysates were centrifuged at 13,000 rpm at 4 °C for 20 min. The supernatant was collected. For enrichment of glycoproteins in experiments involving α-dystroglycan, 200 µg of protein of the total lysates was mixed with 100 µg of wheat germ agglutinin (WGA) agarose beads (Sigma-Aldrich) as described previously [50]. Equivalent amounts of protein lysates non-incubated or incubated with WGA agarose beads were resolved on 8% SDS–PAGE gels and transferred to PDVF membranes (Millipore). Western blot analysis of equal protein loading was performed with the following primary antibodies: mouse monoclonal anti-α-dystroglycan (IIH6C4) (1:1000; Millipore); mouse monoclonal anti-β-dystroglycan (43DAG1/8D5) (1:500; Novocastra); sheep anti-α-dystroglycan core (317) (1:200; Stephan Kroger laboratory); rabbit monoclonal anti-NOTCH1 (val1744) (1:1000; Cell Signaling); rabbit polyclonal anti-POGLUT1 (1:100; Novus Biologicals) and rabbit polyclonal anti-GAPDH (1:2000; Sigma-Aldrich). Immunoreactivity was detected with secondary antibodies conjugated to horseradish peroxidase (Jackson ImmunoResearch) and developed with SuperSignal West Femto (Thermo Scientific) using an ImageQuant LAS 4000 MiniGold System (GE Healthcare Life Sciences).

Ligand overlay assay was performed as previously described with minor modifications [50,51]. Briefly, PVDF membranes were incubated with Engelbreth-Holm-Swarm laminin (Sigma-Aldrich) overnight at 4 °C in laminin binding buffer.

Then, membranes were washed and incubated with anti-laminin (LAM-89) (1:100; Sigma-Aldrich) primary antibody and the corresponding secondary antibody. Blots were imaged using the protocol described for Western blots.

## Mass spectrometric analysis of mouse NOTCH3 glycosylation

Proteins were transiently expressed in HEK293T cells and purified for mass spectral analysis following the method described previously [52]. Briefly, cells were grown in a 10 cm petri dish with DMEM medium with High glucose, 10% Bovine Calf Serum and 1% penicillin and streptomycin. Cells were transfected with 6 μg plasmid encoding mouse NOTCH3 EGF1–34-Myc-His$_6$ using 24 μg polyethyleneimine (PEI) in 6 mL OPTI-MEM (Invitrogen cat:31985088). Transfection was scaled up when protein was poorly expressed. Medium was collected 3 days later, and protein was purified using Ni-NTA beads (Qiagen, Cat:30230). Purified proteins were reduced, alkylated, and subjected to in-solution digestion with trypsin or chymotrypsin separately. The resulting peptides were analyzed by Q-Exactive Plus Orbitrap mass spectrometer (Thermo Fisher, Waltham, MA, USA) coupled with an Easy nano-LC HPLC system with a C18 EasySpray PepMap RSLC C18 column (50 μm × 15 cm, Thermo Fisher Scientific, Waltham, MA, USA). Peptides modified with glucose and/or fucose were identified by using PMI-Byonic (version 2.10.5; Protein Metrics) as a node in Proteome Discoverer (v2.1). Semi-quantitative Extracted Ion Chromatograms (EICs) of selected ions were generated to compare relative amounts of glucosylated and non-glucosylated glycoforms of each modified peptide (S5 Fig).

## Statistical analysis

Comparison of two means was carried out with unpaired *t* tests. Comparison of multiple means was carried out with one-way, two-way, or three-way ANOVA depending on the number of variables, followed by Tukey's or Šidák's multiple comparisons tests. Prism (GraphPad) was used for statistical analysis. The raw data and statistical analyses used for all graphs are included in S1 Table.

## Supporting information

**S1 Fig.  *Poglut1-cKO* muscles do not show abnormal grip strength when normalized by body weight but exhibit reduced quantal content. A)** Grip test of control and *cKO* mice normalized by body weight. **B)** Electrophysiological recordings from LAL muscles at low frequency stimulation (0.5 Hz, 100 s). **C)** Quantification of the EPP amplitude, miniature EPP (mEPP) amplitude, and the quantal content from control and *Poglut1-cKO* muscles. The mean mEPPs amplitude showed no significant difference between control and cKO fibers, suggesting comparable spontaneous release. While the mean size of the EPPs showed no significant difference between cKO and control, the quantal content values of cKO were significantly lower than that in control, suggesting an alteration in neurotransmission in cKO mice (*Poglut1-Ctr*: 3 mice (52 terminals); *Poglut1-cKO*: 3 mice (33 terminals)). In A and C, each dot represents an animal. Mean±SD is shown. Two-way ANOVA with Šidák's multiple comparisons test (A); Unpaired *t* test (C). Row factor (the overall effect of genotype) is shown for A. ns: not significant, \**P*<0.05.
(TIF)

**S2 Fig.  *Poglut1-cKO* muscles exhibit reduced αDG glycosylation and laminin binding.** Shown is α-DG glycosylation status in *Poglut1-cKO* and control mouse muscles at P0 and P20 assessed by western blotting using antibodies against the glycosylated form of α-DG (αDG-IIH6) and the core α-DG, and by laminin overlay assay. An antibody against β-DG was used to assess loading. Wheat germ agglutinin-enriched muscle lysates from control and cKO mice were used. The western blot showed a reduced expression level of glycosylated α-dystroglycan in cKO mice at P0 and P20, as well as a diminished binding activity in the laminin overlay assay when compared to control mice. Each lane represents an individual animal. Boxes mark samples taken from non-adjacent wells of the same gel.
(TIF)

**S3 Fig. Representative images of myofibers with internal nuclei in TA muscles of *Poglut1-cKO* and sibling control animals.** Shown is the DAPI, laminin co-staining of TA muscles from the indicated genotypes at E16, P0, and P21. Note a higher frequency of myofibers with internal nuclei in P0 and P21 *Poglut1-cKO* muscles compared to sibling controls without *Poglut1* deletion (*Poglut1*^+/+, *Poglut1*^flox/flox, or *Pax7*^Cre; *Poglut1*^+/+). Quantification is shown in Fig 3A.
(TIF)

**S4 Fig. Representative images of myogenic progenitors cultured for five days in proliferation media.** Shown is the PAX7 MYOD Ki67 triple staining of myogenic progenitors isolated from forelimb and hindlimb muscles of P4 *Poglut1-cKO* and sibling controls without *Poglut1* deletion (*Poglut1*^+/+, *Poglut1*^flox/flox, or *Pax7*^Cre; *Poglut1*^+/+) and cultured in proliferation media for five days. TOPRO-3 marks the nuclei. Yellow arrowheads, white arrows and white arrowheads mark examples of quiescent muscle stem cells (PAX7^+ MYOD^− Ki67^−), cycling progenitor cells (PAX7^+ MYOD^+ Ki67^+), and non-cycling precursor cells (PAX7^− MYOD^+ Ki67^−), respectively. Quantifications are shown in Fig 3H.
(TIF)

**S5 Fig. Mass Spectra and Extracted Ion Chromatograms of peptides modified by POGLUT1 from NOTCH3.** Representative MS/MS spectra are shown on the left, and Extracted Ion Chromatograms (EICs) on the right for peptides containing the POGLUT1 consensus site from mouse NOTCH3 EGF repeats. **A)** Peptide with *O*-glucose monosaccharide on EGF2. **B)** Peptide with *O*-glucose-xylose-xylose trisaccharide on EGF3. **C)** Peptide with *O*-glucose-xylose-xylose trisaccharide on EGF9. **D)** Peptide with *O*-glucose-xylose-xylose trisaccharide on EGF11. **E)** Peptide with *O*-glucose-xylose-xylose trisaccharide on EGF13. **F)** Peptide with *O*-glucose-xylose-xylose trisaccharide and *O*-GlcNAc monosaccharide on EGF15. **G)** Peptide with *O*-glucose-xylose-xylose trisaccharide and *O*-GlcNAc monosaccharide on EGF16. **H)** Peptide with *O*-glucose-xylose-xylose trisaccharide and *O*-fucose-HexNAc-Hexose trisaccharide on EGF17. **I)** Peptide with *O*-glucose-xylose-xylose trisaccharide and *O*-fucose monosaccharide on EGF19. **J)** Peptide with *O*-glucose-xylose-xylose trisaccharide and *O*-fucose monosaccharide on EGF23. **K)** Peptide with *O*-glucose monosaccharide on EGF25. **L)** Peptide with *O*-glucose-xylose-xylose trisaccharide and *O*-fucose monosaccharide on EGF29. **M)** Peptide with *O*-glucose-xylose-xylose trisaccharide on EGF32. The b- and y-ions are indicated in blue and red font, respectively. ~ indicates ions that lost glucose in the gas phase during fragmentation. Due to the lability of *O*-glycans in the gas phase during collision, these modifications can fall off which can lead to incorrect annotation of the modified amino acid. However, we can predict the correct position based on the putative consensus sequence for modification by POGLUT1.
(PDF)

**S1 Table. Raw numerical values and summary statistics for graphs shown in the paper.**
(XLSX)

## Acknowledgments

We thank Mario Lopez for excellent technical assistance, Joshua Adams for discussions at the early stage of the project, Stephan Kroger for the gift of anti-α-dystroglycan core antibody, Lucía Tabares for permission to use their electrophysiology setup, the Microscopy Core facility of the Baylor College of Medicine Intellectual and Developmental Disabilities Research Center (IDDRC), and Ashutosh Pandey for advice on data analysis.

## Author contributions

**Conceptualization:** Soomin Cho, Emilia Servián-Morilla, Victoria Navarro, Carmen Paradas, Hamed Jafar-Nejad.

**Formal analysis:** Soomin Cho, Emilia Servián-Morilla, Victoria Navarro, Youxi Yuan, Robert S Haltiwanger, Carmen Paradas, Hamed Jafar-Nejad.

**Funding acquisition:** Emilia Servián-Morilla, Radbod Darabi, Carmen Paradas, Hamed Jafar-Nejad.

**Investigation:** Soomin Cho, Emilia Servián-Morilla, Victoria Navarro, Beatriz Rodriguez-Gonzalez, Raquel Cano, Arjun A. Rambhiya.

**Methodology:** Soomin Cho, Emilia Servián-Morilla, Victoria Navarro, Youxi Yuan, Robert S Haltiwanger, Carmen Paradas, Hamed Jafar-Nejad.

**Project administration:** Hamed Jafar-Nejad.

**Supervision:** Robert S Haltiwanger, Carmen Paradas, Hamed Jafar-Nejad.

**Validation:** Soomin Cho, Emilia Servián-Morilla, Victoria Navarro.

**Visualization:** Soomin Cho, Emilia Servián-Morilla, Victoria Navarro, Youxi Yuan.

**Writing – original draft:** Soomin Cho, Emilia Servián-Morilla, Victoria Navarro, Carmen Paradas, Hamed Jafar-Nejad.

**Writing – review & editing:** Soomin Cho, Emilia Servián-Morilla, Victoria Navarro, Youxi Yuan, Radbod Darabi, Robert S Haltiwanger, Carmen Paradas, Hamed Jafar-Nejad.

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
