## [Decision Letter · Decision Letter 0]

29 Jun 2025

PGENETICS-D-25-00545

The glycosyltransferase POGLUT1 regulates muscle stem cell development and maintenance in mice

PLOS Genetics

Dear Dr. Jafar-Nejad,

Thank you for submitting your manuscript to PLOS Genetics. After careful consideration, we feel that it has merit but does not fully meet PLOS Genetics's publication criteria as it currently stands. Therefore, we invite you to submit a revised version of the manuscript that addresses the points raised during the review process.

Please submit your revised manuscript within 30 days Jul 29 2025 11:59PM. If you will need more time than this to complete your revisions, please reply to this message or contact the journal office at plosgenetics@plos.org. Please include the following items when submitting your revised manuscript:

We look forward to receiving your revised manuscript.

Kind regards,

Ophir Klein

Academic Editor

PLOS Genetics

Giovanni Bosco

Section Editor

PLOS Genetics

Aimée Dudley

Editor-in-Chief

PLOS Genetics

Anne Goriely

Editor-in-Chief

PLOS Genetics

**Journal Requirements:**

https://journals.plos.org/plosgenetics/s/submission-guidelines#loc-parts-of-a-submission

- ® on page: 19

- TM on pages: 19, and 20.

Potential Copyright Issues:

i) Please confirm (a) that you are the photographer of 1A, or (b) provide written permission from the photographer to publish the photo(s) under our CC BY 4.0 license.

6) We note that your Data Availability Statement is currently as follows: "All relevant data are within the manuscript and its Supporting Information files.". Please confirm at this time whether or not your submission contains all raw data required to replicate the results of your study. Authors must share the “minimal data set” for their submission. PLOS defines the minimal data set to consist of the data required to replicate all study findings reported in the article, as well as related metadata and methods (https://journals.plos.org/plosone/s/data-availability#loc-minimal-data-set-definition).

7) Please amend your detailed Financial Disclosure statement. This is published with the article. It must therefore be completed in full sentences and contain the exact wording you wish to be published.

2) If any authors received a salary from any of your funders, please state which authors and which funders..

8) Please ensure that the funders and grant numbers match between the Financial Disclosure field and the Funding Information tab in your submission form. Note that the funders must be provided in the same order in both places as well.  

9) Please ensure that the affiliations of the authors listed on the manuscript title page (CITY) do exactly match with the affiliations provided in the online submission form

NOTE: Affiliations should include a department (if applicable), an institution, a CITY, and a country. 

Please indicate by return email the full and correct funding information for your study and confirm the order in which funding contributions should appear.

**Reviewers' comments:**

Reviewer's Responses to Questions

**Comments to the Authors:**

Reviewer #1: The authors have addressed all my concerns. Congratulations!

Reviewer #2: The Authors have addressed or explained most of my queries. My only comments are the following:

1. The Authors choose to use controls which are siblings but are of mixed genotypes, which I am not sure of. How are the Authors confident that all control genotypes (Poglut1+/+, Poglut1flox/flox, or Pax7Cre; Poglut1+/+) are identical?

2. How many biological replicates have been used for the P21 timepoint in Figure 1B and C? A minimum of 3 replicates are required for drawing conclusions from statistical tests.

3. The grip strength data normalized to body weight should be included in the manuscript.

**Have all data underlying the figures and results presented in the manuscript been provided?**

Reviewer #1: Yes

Reviewer #2: Yes

PLOS authors have the option to publish the peer review history of their article (what does this mean? ). If published, this will include your full peer review and any attached files.

**Do you want your identity to be public for this peer review?** For information about this choice, including consent withdrawal, please see our Privacy Policy .

Reviewer #1: No

Reviewer #2: **Yes: ** Sam J Mathew

**Figure resubmission:**
---

## [Editor Report · Decision Letter 1]

14 Jul 2025

Dear Dr Jafar-Nejad,

We are pleased to inform you that your manuscript entitled "The glycosyltransferase POGLUT1 regulates muscle stem cell development and maintenance in mice" has been editorially accepted for publication in PLOS Genetics. Congratulations!

Yours sincerely,

Ophir Klein

Academic Editor

PLOS Genetics

Giovanni Bosco

Section Editor

PLOS Genetics

Aimée Dudley

Editor-in-Chief

PLOS Genetics

Anne Goriely

Editor-in-Chief

PLOS Genetics

Comments from the reviewers (if applicable):

**Data Deposition**

http://datadryad.org/submit?journalID=pgenetics&manu=PGENETICS-D-25-00545R1

**Press Queries**

---

## [Editor Report · Acceptance letter]

PGENETICS-D-25-00545R1

The glycosyltransferase POGLUT1 regulates muscle stem cell development and maintenance in mice

Dear Dr Jafar-Nejad,

We are pleased to inform you that your manuscript entitled " 

The glycosyltransferase POGLUT1 regulates muscle stem cell development and maintenance in mice" has been formally accepted for publication in PLOS Genetics! Your manuscript is now with our production department and you will be notified of the publication date in due course.

With kind regards,

Zsofia Freund

PLOS Genetics

On behalf of:
